# Phytomanagement of a Lead-Polluted Shooting Range Using an Aromatic Plant Species and Its Effects on the Rhizosphere Bacterial Diversity and Essential Oil Production

**DOI:** 10.3390/plants11223024

**Published:** 2022-11-09

**Authors:** Anabel Saran, Lucia Fernandez, Cinthia Yanela Latini, Monica Bellozas Reinhard, Marisol Minig, Sofie Thijs, Jaco Vangronsveld, Luciano Jose Merini

**Affiliations:** 1Scientific Research Agency, CONICET, Santa Rosa L6300, La Pampa, Argentina; 2EEA-INTA Anguil, CONICET, Anguil L6326, La Pampa, Argentina; 3INCITAP-UNLPam, CONICET, Santa Rosa L6300, La Pampa, Argentina; 4Department of Chemistry, National University of La Pampa, Santa Rosa L6300, La Pampa, Argentina; 5Environmental Biology, Centre for Environmental Sciences, Hasselt University, Agoralaan Building D, 3590 Diepenbeek, Belgium; 6Department of Plant Physiology and Biophysics, Faculty of Biology and Biotechnology, Maria Curie Sklodowska University, Akademicka, 19, 20-400 Lublin, Poland

**Keywords:** lead, field trial, phytostabilization, phytomanagement, aromatic plants, bacterial community

## Abstract

This field study aimed to assess the baseline conditions of a long-term shooting range in Argentina polluted with 428 mg kg^−1^ lead (Pb) to evaluate the establishment and development of *Helianthus petiolaris* plants and address the efficacy of the phytomanagement strategy through: (i) element accumulation in plant tissues; (ii) rhizosphere bacterial diversity changes by Illumina Miseq™, and (iii) floral water and essential oil yield, composition, and element concentration by GC–MS and ICP. After one life cycle growing in the polluted sites, in the roots of *Helianthus petiolaris* plants, Pb concentration was between 195 and 304 mg kg^−1^ Pb. Only a limited fraction of the Pb was translocated to the aerial parts. The predominance of the genus *Serratia* in the rhizosphere of *Helianthus petiolaris* plants cultivated in the polluted sites and the decrease in the essential oil yield were some effects significantly associated with soil Pb concentration. No detectable Pb concentration was found in the floral water and essential oil obtained. Extractable Pb concentration in the soil reduced between 28% and 45% after the harvest.

## 1. Introduction

Shooting range areas are the second largest source of Pb pollution after industry, ranging from 10 to 60,000 tons of annual depositions in different countries [1]. Elements can leach from bullets and bullet fragments by chemical processes, such as oxidation, carbonation, and hydration. Soil pH, the weathering actions of air, water, organic acids, and microbial activity play an important role in the transformation of metallic Pb to secondary Pb minerals [2]. Either way, these processes result in elevated concentrations of soluble trace elements in the soil, ground, and surface water [3]. A bullet primarily consists of 90% Pb, 2–7% Sb, 0.5–2% As, 0.5% Ni, and traces of Ag [4]. Several studies have reported Pb pollution in shooting range soils exceeding the US EPA threshold of 400 mg kg^−1^ of Pb (USEPA 1996) [5] or even total concentrations of 10,000 mg kg^−1^ [6,7].

Elevated concentrations of elements such as Pb in soils can have negative effects on soil microbial communities, plants, animals, and human health [8]. In addition, cereal products, grains, and vegetables (especially potatoes and leafy vegetables) are the most important contributors to Pb dietary exposure in the general European population [9]. In this context, the transfer of Pb into the food chain is a risk that must be taken into consideration [10], as many shooting range areas are used as meadowland after being decommissioned or during nonshooting periods, and some are even used for the production of food crops [11,12].

A promising approach for the economic valorization and phytomanagement of such areas is to grow aromatic plants on them. Since these are nonfood crops, the risk of food chain contamination is lower. In this way, fast-growing aromatic plants can be considered for phytostabilization, in combination with the production of high-market-value essential oils and other added-value subproducts, without the risks of trace element cross-contamination of the products [13]. Plants specially selected for phytomanagement can cope with soil pollutants and, at the same time, increase soil function and fertility with inputs of organic matter, atmospheric CO_2_ fixation, and enhancement of local biodiversity [14]. Despite a growing track record of field success [15,16,17,18], more field studies are needed to prove that phytomanagement is effective in order to rigorously measure its underlying economics and to expand its applications.

The uptake of trace elements depends on not only the mobility and bioavailability of the elements in the soil and the expression of element transporters and detoxification genes by the plant, but also the plant-associated microbiota [19]. Indigenous rhizosphere microflora may increase plant element uptake by redox transformations, dissolution, leaching, or immobilization through organic molecule-trace element binding and precipitation [20] and can also enhance plant tolerance to trace element stress and biomass production [21]. Improving knowledge of the complex interactions mediated by plant rhizosphere bacteria can lead to nature-based solutions to improve the performance of these plants when grown in soils contaminated with trace elements.

*Helianthus petiolaris* Nutt. is an annual plant species, considered the wild ancestor of *H. annuus* [22]. This species grows in xeric, sandy soils in the center of South America with a flowering period from December to March [23]. The in vitro and greenhouse background of trace element tolerance of this species was first reported by Saran et al. (2019) [24]. The combination of such features and the reported potential of its essential oils to be used as biotechnological pest control in stored grains [25] makes this aromatic plant an outstanding candidate for designing phytomanagement strategies.

This study aimed to assess the baseline conditions of a long-term polluted shooting range in Argentina to evaluate the establishment and development of *H. petiolaris* plants and address the efficacy of the phytomanagement strategy through Pb stabilization, as well as to investigate the effects of the Pb pollution on the rhizosphere bacterial diversity and essential oil yield and composition.

After one life cycle growing in the polluted sites, *H. petiolaris* phytostabilized within the roots a significant Pb fraction, reducing the ammonium acetate EDTA extractable fraction of Pb on the soil, which in turn reduce the dispersion of Pb to the ground and surface waters. Added-value by-products were obtained from its biomass without risks of Pb contamination. The rhizospheric bacterial community appeared to adapt to cope with the presence of Pb in the soil and could be associated with the significantly higher development of plant biomass when this species grows in polluted soil.

## 2. Results

Two sites were selected inside a shooting range to perform a vegetation field experiment (sites 1 and 2), and one nonpolluted site outside the shooting range was used as negative control (site 3) (Appendix A). The physicochemical properties of the soil at each site and the total and ammonium acetate EDTA extractable Pb concentration are presented in Table 1. The predominant texture at the polluted sites was sand. The pH of the soils ranged between 7.4 and 7.7, which can be considered adequate for the healthy growth and development of plants. Optimal electric conductivity (EC) levels for plant development should be in the range of 110–570 mS m^−1^; the low EC levels at site 2 indicated low nutrient concentrations. Moreover, the soil at site 2 contained less than half of the organic matter (OM) than the other sites (0.5%). The levels of pollution were not significantly different on sites 1 and 2 in terms of total Pb concentrations. However, ammonium acetate EDTA extractable Pb on site 2 was higher. Furthermore, the ammonium acetate EDTA extractable Pb content was lower in the lower layer (15–30 cm) of the polluted sites than in the top layer.

After transplantation, plant adaptation was monitored during the first weeks. Half of the plants transplanted to site 2 did not survive, which might be related to the unfavorable combination of low OM content, low EC, and high Pb availability (measured as ammonium acetate EDTA extractable fraction). However, those plants able to survive and grow on both, polluted sites 1 and 2, developed significantly higher (*p* ≤ 0.05) fresh weights compared with the plants grown on the nonpolluted control site (Figure 1). Growth rate and fresh weight are unrelated among sites due to the branching in the stems of *H. petiolaris*, which makes it possible to find shorter plants with higher plant biomass.

After 6 months of growth in the polluted sites 1 and 2, in the roots of *H. petiolaris* plants, Pb concentrations were between 195 and 304 mg kg^−1^ dry weight (Figure 1). Between 23 and 27 mg kg^−1^ of Pb was translocated to the shoots, and between 4 and 10.5 mg kg^−1^ was translocated to the flower head. No significant differences were found between the concentrations accumulated in the different plant organs and the polluted site where the plants were cultivated. The highest bioaccumulation factor (BAF) was recorded in the roots. Furthermore, although the nonpolluted control site contained 15 times lower Pb concentration, the BAF was not significantly different regarding those plants cultivated in the polluted sites (Table 2). Pb was not easily transported to shoots, as indicated by translocation factor (TF) values <1. However, the amount of Pb sequestrated within the roots decreased the ammonium acetate EDTA extractable Pb fraction by 28–45% in the upper soil layer (Table 1).

The soil physicochemical properties of site 2 did not allow extraction of pure DNA using the MOBIO PowerSoil Isolation Kit. Hence, it was only possible to investigate the bacterial communities of plants growing on the polluted site 1 and the nonpolluted site (control site 3). The Chao microbial richness parameter and the Shannon microbial diversity parameter were not significantly different between samples from polluted site 1 and the unpolluted control site (Figure 2a). A total of 35,770 high-quality sequences were obtained from Illumina MiSeq sequencing, and 274 ASVs were obtained across all libraries from the rhizosphere of *H. petiolaris* plants grown on the polluted and nonpolluted site (Figure 2). The genus *Streptomyces* was common to both sites. On the other hand, the genus *Serratia* was restricted to plants grown in polluted soils, and the genera *Nitrobacter* and *Herbaspirillum* were only found in the rhizosphere of the plants growing on the nonpolluted site.

The composition of the essential oils from *H. petiolaris* plants was similar regardless of the site where they were cultivated (Table 3). However, plants growing on site 1 produced significantly less essential oil than the plants growing on the nonpolluted site, and plants growing on site 2 did not produce essential oil. The Pb concentration in the floral water and essential oil from plants grown on site 1 was below the detection limit, and the Pb concentrations measured in the plant waste after the essential oil extraction were proportional to those measured in the plant tissues after the harvest, showing that Pb is strongly sequestrated in the plant tissue.

## 3. Discussion

Toxicity thresholds of Pb to plants vary over more than two orders of magnitude and increase with decreasing effective cation exchange capacity of the soil [26]. The soil of the polluted sites in our study contained between 317 and 562 mg Pb kg^−1^ (Table 1), exceeding the thresholds established by the Argentinian law (No. 24.585) and the US EPA standards (1996) [5]. However, the toxic effects of trace elements, particularly lead, on ecosystems and human health are primarily connected with the easily available forms, which often are only a fraction of the total contents [27]. In the shooting range sites selected in our study, the ammonium acetate EDTA extractable concentration of Pb was between 13 and 28 times lower than the total concentration (Table 1). Furthermore, significantly higher concentrations of ammonium acetate EDTA extractable Pb were found in the upper and lower soil layers of site 2. The soil of this site had lower OM levels (Table 1), which might explain the higher Pb extractability by ammonium acetate EDTA. At both polluted sites, the Pb content was higher in the upper (0–15 cm) than the lower (15–30 cm) layer, indicating a slow vertical percolation of the pollutants. This is consistent with recent findings at shooting ranges on peatlands, where the concentrations of Pb, Cu, and Sb were found to decrease sharply in the vertical profile [28]. In this scenario of Pb pollution, of sandy and poor soils, with low metal availability and reduced leaching, phytomanagement can be an environmentally responsible approach. After 6 months of growth on the polluted site, *H. petiolaris* plants showed different growth performances in the function of the site (Figure 1). The significant differences in soil characteristics of the two polluted sites evidently had straight effects on plant growth and development, restricting the survival rate after transplanting. Similar results were reported by Ghazaryan et al. (2019) [29], who investigated the copper phytostabilization potential of wild plant species growing in different mine-polluted areas. In this way, although the logistics in full scale projects must be considered, such obstacles could be solved by plant replacement.

The low TF values (Table 2) obtained indicate that only limited amounts of Pb are transferred to the aboveground parts of these plants, minimizing the risks of Pb entering into the food chain. Kiran et al. (2017) [30] reported that a higher percentage of Pb tends to get restricted on and within the cell wall complex of the roots, while only a limited fraction is translocated to the aerial parts of several plant species. Furthermore, if we compare the results obtained in the field trial with the results obtained in in vitro and in greenhouse previous studies [26], higher BAF values were obtained in the field. This might be due to the longer growth period (6 months) before the harvest and the unlimited development of the roots.

According to Li et al. (2019) [31], trace elements in soils can adversely affect the numbers, abundance, and activity of microorganisms. In our study, we did not observe differences in the numbers and abundance of rhizosphere microorganisms present on the polluted and unpolluted sites (Figure 2a), as many other studies carried out on polluted sites [32,33]. Płociniczak et al. (2018) [21] highlighted in their study that the bacterial community structure of rhizosphere soils depended more on the plant than on the distance and metal concentrations in the soil.

A predominance of the genus *Serratia* was found in the rhizosphere of *H. petiolaris* plants grown on the polluted site (Figure 2b). This genus was previously isolated from a flowing stream polluted with trace elements [34] and from industrial polluted waters and mining affected areas [11]. Singh and Jha (2016) [35] demonstrated the plant-growth-promoting (PGP) potential of the *Serratia marcescens* CDP-13 strain isolated from *Capparis decidua* and described its ability to protect plants from the deleterious effect of biotic and abiotic stressors. *H. petiolaris* plats colonized by *Serratia* developed significantly higher (*p* ≤ 0.05) fresh weights compared with the plants grown on the nonpolluted control site (Figure 1). This can be related to the PGP capacity of this strain. Therefore, future investigations on the bioaugmentation and distribution of the *Serratia* taxa may provide more information for improving the survival of plants in Pb-polluted soils.

On the other hand, Rotkittikhun et al. (2010) [36] reported that Pb enhanced the oil content of vetiver (*Chrysopogon zizanioides*) on Pb-polluted mining sites. In our study, plants grown on the Pb-polluted sites delivered significantly fewer essential oils than plants cultivated on the nonpolluted site (Table 3). The decrease in essential oil yield, especially in site 2, where essential oil was not produced, can be significantly associated with soil Pb available concentration and soil physicochemical properties. However, the general composition of the essential oils obtained from plants on the polluted site 1 was similar to that obtained from plants growing on the nonpolluted site. This agrees with the findings of Khajanchi et al. (2013) [37], who also not found significant changes in oil constituents after growing lemon grass plants on polluted sites. However, it would be valuable to study the traces of the organic compounds.

Although it is necessary to perform more research to elucidate whether this plant species can be used in soils with higher pollution levels, its features make it a promising candidate for the phytostabilization and phytomanagement of trace-element-polluted soils, especially when they are not suitable for producing other commercial food and feed crops. In addition, after oil extraction, biomass waste of aromatic plants containing sequestered trace elements can be used for onsite composting as organic manure or can be incinerated or managed as sanitary hazard waste if the aim is to completely remove the Pb from the site.

## 4. Materials and Methods

### 4.1. Soil Sampling and Site Characterization

Two abandoned shooting range areas in La Pampa, Argentina (site 1: 36° 37′ 11.302′’ S, 64° 15′ 25.366′’ W; site 2: 36° 37′ 11.543′’ S, 64° 15′ 26.884′’ W) were selected to perform this field trial. A nonpolluted area outside the shooting range (0.5 km distance) was used as a control site (36° 37′ 9.264′’ S, 64° 15′ 27.295′’ W) (Appendix A). Soil sampling was performed before the cultivation of *H. petiolaris* and after harvesting. Considering Pb dynamics in soil, samples from horizon A were collected at two different levels (0–15 and 15–30 cm). Six samples were collected per site and horizon. Soil samples were air-dried, crushed, homogenized, and sieved through a 2 mm sieve for physicochemical analysis. Total trace element concentrations were determined using the US EPA 3051 microwave-assisted digestion method with HNO_3_ [5]. Blanks (only HNO_3_) and standard reference (RM No. 143) trace elements in a sewage-sludge-amended soil (Commission of the European Communities) were included. The ‘potentially available’ or ‘labile’ pool of Pb was determined using 0.05 M ammonium acetate EDTA extraction according to Chrastný et al. (2008) [38]. The Pb concentrations in the extracts were determined by inductively coupled plasma-atomic emission spectrometry (ICP-OES, Agilent Technologies, 700 series, Belgium). Sb, As, and Ni were not measured since they are found in very low traces in bullets.

### 4.2. Trial Design

Based on previous research, the recently described aromatic *H. petiolaris* was selected [24,25]. The phytomanagement field trial was conducted according to Yang et al. (2010) [39], and 60 seedlings of *H. petiolaris* were raised in a greenhouse for a month and then transplanted into the experimental sites. A 1 m^2^ grid was regularly outlined across the 50 m^2^ site 1 and site 2, following the top–bottom slope. On this grid, seedlings were planted at 1 m intervals, along with the slope, covering the remediation sites. Before transplanting 30 seedlings in each site, 250 g m^−2^ of hydrous polyaspartic acid hydrogel was applied at the bottom of each hole and gently covered with a thin layer of soil, as a water-retaining agent. Once transplanted, plants were allowed to grow under the local environmental conditions, without further fertilization or irrigation for one full life cycle (6 months). Climate parameters in the Pampa region, Argentina, are described in Loponte and Corriale’s (2019) research [40].

### 4.3. Plant Survival, Biomass, and Trace Element Accumulation

Plant survival, biomass development, and vegetation cover were weekly monitored during the field trial. After 6 months, when reaching the flowering phase, plants were harvested. Roots were washed three times with sterile water to remove any soil particles. Shoots, flowers, and roots were oven-dried (60 °C for 1 week), weighed, digested with 70% HNO_3_ in a heat block, and dissolved in 5 mL of 2% HCl, according to the US EPA 3050B Acid Digestion of Sediments, Sludges, and Soils protocol [41]. Trace element concentrations were determined using inductively coupled plasma-atomic emission spectrometry (ICP-OES, Agilent Technologies, 700 series, Belgium). Blanks (only HNO_3_) and standard references (NIST Spinach 1570a) were included.

The BAF was calculated according to Rafati (2011) [42] using the ratio of element concentration in the plant roots to the soil for the “BAF roots”, the ratio of total element concentration in plant shoots (stem + leaves) to the soil for the “BAF shoots”, and the ratio of element concentration in the plant flower heads to the soil for the “BAF flower heads”. The TF was calculated according to Padmavathiamma and Li (2007) [43] using the ratio of element concentrations in the areal parts of the plants (shoots + flower heads) to the roots.

### 4.4. H. petiolaris Rhizosphere Bacterial Community Analyses

The effects of Pb soil pollution on the rhizosphere bacterial community (soil immediately surrounding the root) of *H. petiolaris* plants were investigated. From each site, five samples (1 g of rhizosphere soil) were collected. Roots were gently shaken to remove the soil weakly attached to them, and the soil obtained was sieved and homogenized by quartering. DNA extraction was performed using the MO BIO PowerSoil Isolation Kit (MO BIO) (Appendix A). Extracted DNA was subjected to two polymerase chain reactions (PCRs) using an Illumina (2014) Nextera XT DNA Library Preparation Kit Data Sheet according to Bentley et al. (2008) [44]. In the first PCR reaction, the primers 515F (5’-GTGCCAGCMGCCGCGGTAA-3′) and 806R (5’-GGACTACHVHHHTWTCTAAT-3′) were used to amplify the 16S rRNA gene (region V4). Bacterial amplicons produced by the first PCR reaction were purified through the NucleoMag^®^ NGS Clean-up and Size Select DNA purification kit (Macherey-Nagel GmbH & Co., Düren, Germany), and 5 μL was used in the second PCR. For multiplexed sequencing, in the second PCR, a sample-specific 10 bp barcode (MID) was attached to the forward primer, followed by the key and a Lib-L Adapters N and S combination sequence (Illumina, San Diego, USA). The bacterial amplicons produced by the second PCR were purified, and the concentration of purified DNA was determined with the Qubit^®^ dsDNA HS Assay Kit (Life Technologies Europe, Ghent, Belgium), according to the manufacturer’s protocol. Equimolar mixtures of different samples were prepared. Sequencing was carried out using 2 × 300 bp paired-end amplicon libraries on a Miseq™ (Illumina) platform following the manufacturer’s instructions.

The 16S rRNA reads obtained by Miseq™ were processed and analyzed using version 1.12 of the DADA2 pipeline in the R project [45]. Sequences were clustered into an amplicon sequence variant (ASV), which records the number of times each exact amplicon sequence variant was observed in each sample. After the clustering, the sequences were aligned and taxonomically classified using the SILVA 132 reference database [46].

### 4.5. Essential Oil Extraction, GC–MS Analysis, and Pb Concentration

The aerial parts of *H. petiolaris* plants were collected after a cultivation period of 6 months. The essential oil was extracted by steam distillation from the dried plant biomass using a benchtop scale extractor (Figmay SRL, Argentina). From each site, five samples of floral water (a by-product from distillation) and essential oil were filtered (0.22 µm), and five samples of the plant waste were digested, as mentioned above, to determine Pb concentrations by inductively coupled plasma-atomic emission spectrometry (ICP-OES, Agilent Technologies, 700 series, Belgium).

The *H. petiolaris* essential oil composition was analyzed in an HP 6890N Series Plus gas chromatograph (Agilent Technologies, Palo Alto, California, EE. UU.), equipped with a model 5973N mass selective detector (Agilent Technologies, Palo Alto, California, EE. UU.) and an HP 6890 Series autoinjector. The separation of the compounds was achieved using an HP-5 MS capillary column (30 m × 0.25 mm I.D., 0.25 m film thickness, and 5% phenylmethylsiloxane), supplied by J&W Scientific (Folsom, CA, USA). Based on a mass scan range of 50–550 atomic mass units (AMU) with SCAN mode, the retention times of the compounds were determined by comparing the MS fragmentation pattern of the standards and the National Institute of Standards and Technology (NIST) 2.0 GC–MS library.

### 4.6. Statistical Analysis

Data were analyzed by using analysis of variance (ANOVA). When ANOVA showed a treatment effect, the Tukey post hoc test was applied to make comparisons between the means at *p* < 0.05. Illumina reads were subjected to Shannon and Chao index analysis in the 3.6.1 version of the R project (R Foundation for Statistical Computing, Vienna, Austria).

## Figures and Tables

**Figure 1 plants-11-03024-f001:**
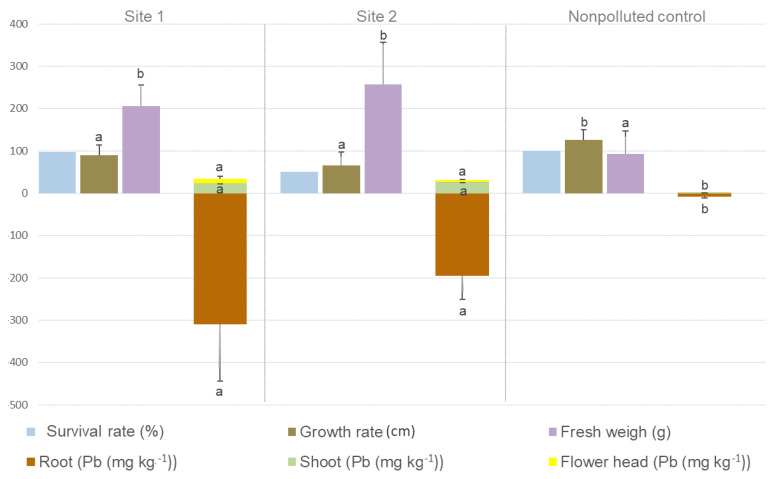
Survival rate (%), growth rate (cm), total fresh weight (g), and concentration of Pb (mg kg^−1^) in roots, leaves, and flower heads of *H. petiolaris* plants growing on the Pb-polluted sites (1 and 2) and the nonpolluted control site. Error bars are S.E. (*n* = 6). Values in the lines followed by the same letter and color are not significantly different at *p* ≤ 0.05 by ANOVA and Tukey test.

**Figure 2 plants-11-03024-f002:**
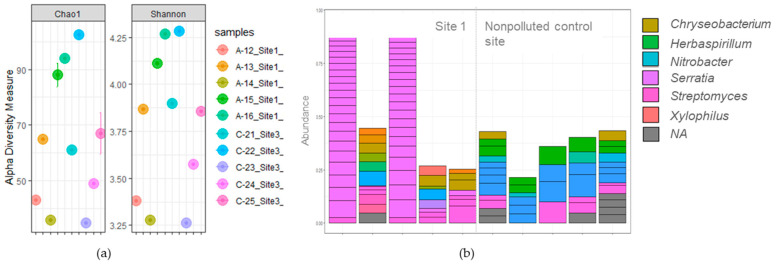
(**a**) Shannon index and Chao diversity analysis of the Illumina Miseq™ reads obtained from the rhizoplane samples of *H. petiolaris* plants growing on the polluted site 1 and the unpolluted control site 3. Five replicates per site are represented in separate circles. (**b**) The abundance of the most abundant genus-level ASVs from a total of 274 genus-level ASVs detected. Five replicates per site are represented in separate bars.

**Table 1 plants-11-03024-t001:** Physicochemical properties, ammonium acetate EDTA extractable, and total lead (Pb) concentrations (mg kg^−1^) of the experimental sites before cultivation and after the harvest.

	Organic Matter (%)	Texture	EC (mS/m)	pH	Total Pb	Ammonium Acetate EDTA Extractable Upper ^a^ Pb	Ammonium Acetate EDTA Extractable Bottom ^b^ Pb	Ammonium Acetate EDTA Extractable Pb after Harvest
Site 1	1.8	Loamy sand	271 ± 5 ^a^	7.7 ± 0.7 ^a^	427.8 ± 135.5 ^a^	15.1 ± 5.1 ^b^	7.3 ± 4.0 ^b^	10.4 ± 6.8 ^a^
Site 2	0.5	Sandy	18 ± 6 ^b^	7.4 ± 0.3 ^a^	416.5 ± 102.1 ^a^	31.1 ± 8.3 ^a^	23.6 ± 9.9 ^a^	24.2 ± 13.5 ^a^
Nonpolluted	1.9	Loamy sand	200 ± 4 ^a^	7.4 ± 0.5 ^a^	20.5 ± 3.4 ^b^	3.9 ± 2.1 ^c^	4.3 ± 1.4 ^b^	2.8 ± 6.8 ^b^

Values are mean ± SE (*n* = 6); <DL: below the detection limit (0.05 mg/kg); EC: electric conductivity; ^a^ (0–15 cm); ^b^ (15–30 cm). Values in a column followed by the same letter are not significantly different at *p* ≤ 0.05 by ANOVA and Tukey test. Reference material (RM No. 143) trace element recoveries were between 85% and 110%.

**Table 2 plants-11-03024-t002:** Pb bioaccumulation factors (BAF) in roots, shoots, and flower heads and translocation factors (TF) of *H. petiolaris* plants growing in the experimental sites.

	BAF			TF
	Roots	Shoots	Flower	
Site 1	0.72 ± 0.35 ^a^	0.05 ± 0.03 ^a^	0.02 ± 0.00 ^a^	0.27 ± 0.63 ^a^
Site 2	0.47 ± 0.12 ^a^	0.06 ± 0.05 ^a^	0.01 ± 0.00 ^a^	0.23 ± 0.19 ^a^
Nonpolluted	0.42 ± 0.22 ^a^	0.09 ± 0.03 ^a^	-	0.21 ± 0.18 ^b^

Values are mean ± S.E. (*n* = 6); values in a column followed by the same letter are not significantly different at *p* ≤ 0.05 by ANOVA and Tukey test.

**Table 3 plants-11-03024-t003:** Identification of compounds present in *H. petiolaris* essential oils obtained from the Pb-polluted site 1 and the nonpolluted control site.

Rt (min)	Compounds	CAS	Relative Area	
			Nonpolluted	Site 1
7.84	α-Pinene	7785-70-8	66	67
8.25	Camphene	79-92-5	8	8
18.53	Acetic acid,1,7,7-trimethyl-bicyclo[2,1,1]hept-2-yl ester	92618-89-8	10	10
23.87	D-germacrene	23986-74-5	16	15
9.04	Thujene	3387-41-5	<1	<1
9.12	β-Pinene	127-91-3	<1	<1
10.77	Limoneno	5989-54-8	<1	<1
12.92	α-Pinene epoxide	1686-14-2	<1	<1
21.45	β-Cubene	13744-15-5	<1	<1
22.24	Caryophyllene	87-44-5	<1	<1
22.63	α-Bergamotene	17699-05-7	<1	<1
23.14	Humulene	6753-98-6	<1	<1
26.41	Caryophyllene oxide	1139-30-6	<1	<1
Oil yield			0.14	0.014
mg Pb/L floral water			<dL	<dL
mg Pb/L essential oil			<dL	<dL
mg Pb/Kg plant waste			2.18 ± 0.20	23.57 ± 0.7

Values are expressed as relative percentages. Rt: retention time. <DL: below the detection limit (0.05 mg/L). Values are mean ± S.E. (*n* = 5).

## Data Availability

Not applicable.

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
