# Peer review of "Phytomanagement of a Lead-Polluted Shooting Range Using an Aromatic Plant Species and Its Effects on the Rhizosphere Bacterial Diversity and Essential Oil Production"

_plants, 2022, doi:10.3390/plants11223024_

Round 1

Reviewer 1 Report

Dear Authors,

the subject of your article is very important from the point of view of environmental protection. The research results can be used for phytoremediation of shooting ranges (especially large military areas). I believe that the work can be accepted for publication. However, I would like to ask you to respond to the following issues:

1. line 38: “Shooting ranges areas are second large source of Pb pollution” - and what is the first and most important source of pollution?

2. line 64: please correct [15-18];

3. In the case of the tested plants, we are dealing with phytostabilization, which does not clean the area, but only immobilizes Pb until the plants decompose. Can the authors suggest a technology for removing lead retained in the roots from the ground?

4. line 100: “…at the polluted sites was sand (or sandy soil) – please correct;

5. Table 1: detection limit is usually DL, not dl; please add DL value in the table; what do the letters "a" and "b" mean in this table - soil layer or significance?

6. line 122: please correct kg-1;

7. line 132: please correct “The highest bioaccumulation factor (BAF) was….”;

8. line 135: please correct “…as indicated by translocation factor (TF)…”;

9. lines 278, 282, Table 2: please standardize the notation of BAF and TF (either capital or small letters – I would suggest the latter);

10. Figure S2: illegible drawing; symbols in the legend do not match the symbols in the picture (different colors – red, black); maybe you should apply different colors to the different symbols;

11. Table 3: detection limit DL;

12. line 170: and above 300 mg/kg Pb is not toxic? Why? Please rewrite this sentence;

13. lines 179-180: please explain, why higher extractability of ions decreases EC?

14. lines 181-184: it is obvious because Pb and Cu show affinity for soil organic matter (and SOM accumulates mainly in the upper layer)

15. References: [11]=[36]; [21]=[34] – please remove and correct in the text

Author Response

Dear Reviewer,

First of all, thank you for your time and efforts to review this manuscript. We truly appreciate the feedback which indeed helped us to improve the quality of the paper. As required, the corrections in the manuscript have been amended to address all the points raised (see changes in track changes mode). Figure S2 was deleted as was not possible to modify the symbols chosen by the program. We have also added at the end of the discussion the possible technologies used for removing lead retained in the roots from the site. Attached we provide a detailed answer to each comments.

Reviewer 2 Report

Dear Authors, this manuscript is very interesting and current, however the authors must clarify: 1- it is not clear why the study is limited to Pb only and not to other equally dangerous heavy metals. In this regard, reasons are required. 2- It would be appropriate to illustrate the GC-MS traces of the organic compounds sought and to evaluate whether the consortia of bacteria listed have given rise to bioremediation (e.g. with oxidation actions, etc.) which would add to the phytoremediation zone for the plant of Helianthus petiolaris. Best Regards.

Author Response

Dear Reviewer,

Thank you very much for the comments and suggestions. We have modified the manuscript according them. Was added in Line 255 the reasons of just measure Pb in soil. We have also added in line 228 the importance on study the traces of the organic compounds in the essential oils. Finally, in line 218 was highlighted that future investigations on the bioaugmentation and distribution of Serratia taxa may provide more information for improving survival of plants in Pb polluted soils.

Reviewer 3 Report

Manuscript focuses on phytomanagement of a lead polluted shooting range using an aromatic plant species and its effects on the rhizosphere bacterial diversity and essential oil production. The subject of this manuscript is consistent with the scope of the Journal. These results are important for future of agricultural biotechnological.

However, manuscript can be published in scientific Plants after some changes (major revision):

-           I think Figure 2b should be changed: the generic names of microorganisms be amended in accordance with the taxonomic nomenclature - spelling in italics,

-           Material and methods: In my opinion, very unrepresentative material was taken for analysis. According to the microbiological protocol, there should be at least 5 g of material, as the soil is heterogeneous.

-           Material and methods: Have repeat DNA isolations been performed? Was one set of DNA isolated? Please post the results of the purity tests and the amount of DNA isolated in the supplement.

Author Response

Dear Reviewer,

Thank you for taking the time to review our work.

Figure 2b was improve taking the suggestions provided. In order to clarify Material and Methods, in the section 4.1 was added the steps taken to get root soil samples (Line 293). Also was added in Supplementary Materials the amount of DNA isolated and the purity (Table S1).

Replies to specific comments are given below.

Have repeat DNA isolations been performed?

Two commercial kits and the traditional phenol-chloroform DNA extraction method were tried in advance. The best of them in terms of the best quality DNA obtained was selected.

Was one set of DNA isolated?

Yes, we didn’t repeat the isolation.

Round 2

Reviewer 3 Report

Dear Authors, Thank you for improving the manuscript. I still have a comment about Figure 2b, I do not see any changes unfortunately, perhaps there was a mistake with the inserted Figure.

I have another comment to make to the authors regarding the repetition of DNA isolation. At the present time they are essential for reliable research, please keep this in mind for the future. Then the samples can be combined increasing the variety of results obtained.

Author Response

Dear reviewer,

We have put in Figure 2b the name of the genera in italics but there was a mistake in the insertion of the figure. We have corrected it. Also, in the figures zip folder you can find the figures with better resolution and with the changes suggested.

Thank you also for the suggestion regarding DNA extraction, we will take it into account in our next studies without any doubt.

Sincerely,